# Drink Driving as the Commonest Drug Driving—A Perspective from Europe

**DOI:** 10.3390/ijerph17249521

**Published:** 2020-12-18

**Authors:** Richard Allsop

**Affiliations:** Centre for Transport Studies, University College London, London WC1E 6BT, UK; r.e.allsop@ucl.ac.uk

**Keywords:** drink-driving, social acceptability, blood alcohol concentration (BAC), risk of collision, legal BAC limit, enforcement, penalties, rehabilitation, alcohol interlock, drug driving

## Abstract

People mixing driving motor vehicles with consuming alcohol increases deaths and injuries on the roads, as was established irrefutably in the mid-1960s. This commentary discusses how society across Europe has responded since then to this burden by managing drink driving in the interests of road safety. The principal response has been to set, communicate and enforce limits on the level of alcohol in the blood above which it is illegal to drive and to deal in various ways with drivers found to be exceeding the limits. Achieving reduction in drink-related road deaths has benefitted public health, though the aim to change behaviour of drinking drivers has been a challenge to the profession. Other achievements have included changes in public attitude to drink driving, and reduction in reoffending by convicted offenders through rehabilitation courses and use of the alcohol interlock, which prevents starting of a vehicle by a driver who has drunk too much. There is scope for improved recording of road deaths identified as drink-related, greater understanding of effectiveness in enforcement of the legal limit and improved availability of the alcohol interlock. Relevance of experience with drink driving to management of other drug driving and prospects for building on the achievements so far are discussed.

## 1. Introduction

Across Europe, about 50 people each year per million population are killed in road traffic and about five times as many are seriously injured [1], consumption of alcohol is high by global standards [2], and a widely quoted estimate [3] is that about a quarter of the road deaths are related to drink driving. The more than half a century since evidence from the USA [4] made it irrefutable that drink driving increases the number of road deaths has been a time of growing cohesion among many European countries. So it is interesting to look back on how efforts to manage drink driving have evolved across Europe, take stock of what has been achieved and consider what more might be done. The aim of this commentary is to do just that in respect of legislation, regulation, public information, enforcement, and dealing with convicted drink driving offenders. Drink driving is as much an issue for public health as for traffic law, and so work by health professionals in addressing social, behavioural and medical challenges related to drink driving warrants a counterpart commentary.

In contributing to worldwide efforts to reduce the burden upon individuals and society of premature death and life-changing injury on the roads, our concerns about driver behaviour are often focused upon just how drivers are dealing with the risks that they are encountering minute by minute as they drive. However, when we address the issues of drink driving and other kinds of drug driving our concern about driver behaviour extends backwards in time for some hours, and in the case of heavy drinking many hours, before the driver has taken to the road. In this extended concern we face the challenge of a conflict in terms of road safety between two deep-seated sources of satisfaction and risk in modern life: driving motor vehicles and consuming alcohol or other recreational drugs.

This commentary is based substantially on the author’s involvement in addressing this challenge, beginning with his role in interpreting the game-changing data from the USA in 1964–1965 [5] and continuing until his sharing in pan-European work on drink driving in this century through the European Transport Safety Council (ETSC). It builds upon the European Transport Safety Lecture he gave in October 2016, which is accessible online as part of a videorecording of an event [6] but has not previously appeared in print or as a Powerpoint presentation. The commentary provides only selected references, but readers who seek more comprehensive coverage of the literature can find it in the reports cited here from the ETSC over the last decade drawing upon widespread European expertise including that of the author. 

The commentary begins by demarcating the specific challenge of drink driving within the much wider challenge of alcohol in society and goes on to cite leading examples of work to quantify the effects of alcohol on drivers’ risk of involvement in collisions and thus on the general level of risk to road users. It then describes regulation of drink driving by setting a legal limit to the level of alcohol in the blood while driving: the setting of the limit, its communication to those required to comply with it and its enforcement in order to deter and detect driving while over the limit. It goes on to deal with the treatment of convicted drink driving offenders, including the gradually growing role in Europe of the alcohol interlock in reducing reoffending. Comments on the relevance of this approach to drink driving to managing other forms of drug driving are offered. Concluding sections discuss progress that has been made in addressing the challenge of drink driving, and steps in research and practice that offer the prospect of further progress.

## 2. Alcohol in Society and in Driving

Alcohol can provide pleasure and relief in ways that many people find helpful in enjoying and coping with life, but its use can also cause harm and suffering to users, their associates and society more widely. It does so with a scope and severity that gives rise to far-reaching responses in terms of information, regulation and mitigation that are ongoing subjects of debate and sometimes controversy. This is the wider context in which the specific issue of drink driving and its management is set, and those concerned with drink driving are wise to be aware of and ready to learn from the wider context, but at the same time to keep their efforts focused on reducing that part of the burden of death injury and damage on the roads that would be prevented if drivers drank less before driving or drove less after drinking.

Well before the motor age, being drunk in charge of a vehicle was recognised as being undesirable and made an offence, but research demonstrating the effects on capability of even modest quantities of alcohol also began before the motor age. So by 1960 evidence of adverse effects of modest levels of alcohol upon capability to drive was clear [7] and by this token many drink drivers are far from being drunk. The offence of being drunk in charge of a vehicle, which was in any case hard to enforce, was seen to have only limited relevance to much of drink driving. Scandinavian countries had long experience of having instead set limits to the level of alcohol in the blood at which it was legal to drive, but direct evidence of the effect of modest levels of alcohol on involvement in collisions, and thus on numbers killed or injured and amounts of damage, was still very limited. In the absence of evidence this effect was vigorously debated.

That situation was changed decisively by publication in 1964 [4] of findings of an extensive case-control field study in Grand Rapids, MI, USA, in 1962–1963, soon to be further analysed by the author [5]. These findings were confirmed and refined by a repeat study in Fort Lauderdale, FL, and Long Beach, CA, USA, in 1997–1999 [8]: helped by statistical techniques that had not yet been devised in the mid-1960s, this yielded the graph in Figure 1. This shows how the risk of involvement in a collision, however slight, relative to the risk without alcohol, was estimated to vary with level of alcohol in the blood, measured in g/L, among the populations driving in these cities. 

The strength of evidence from the Grand Rapids study of risk nearly doubling by 0.8–1.0g/L and rising rapidly thereafter proved decisive in influencing North America and much of Europe within a few years to follow the longstanding example of Scandinavian countries in setting limits to the level of alcohol in the blood at which it was legal to drive. Many of the limits set at that time were either 1.0 or 0.8 g/L, and 0.8 g/L is even now the limit in much of North America and the UK.

However, Figure 1 massively understates the harm done by drink driving. It is dominated by variation in the risk of involvement in the large proportion of collisions that result in no more than material damage or minor injury, whereas the currently widely accepted “Safe System” approach [9] to reduction of risk on the road concentrates on reducing the number of collisions resulting in death or life-changing injury. Neither the Grand Rapids nor the Fort Lauderdale and Long Beach studies tried to analyse variation of risk with blood alcohol concentration (BAC) according to severity of collision, and had they done so the numbers of fatal collisions even in their large samples would have been too few to enable the variation with BAC of risk of involvement in a fatal collision to be estimated reliably. This had to await much larger scale assembly of data concerning BACs of drivers in general and of those involved in fatal collisions.

Data of these kinds for Great Britain and the USA enabled analyses leading to the estimates shown in Figure 2 of the variation with BAC of the risk of involvement in a fatal collision. Findings like these in various countries indicate that the risk of involvement in a fatal collision is about doubled at a BAC of 0.3 g/L and multiplied by 5 at 0.5 g/L and by 10 at 0.8 g/L. It is numbers like these that should inform response to the challenge of drink driving, taken with information about how many lives are being lost and life-changing injuries are taking place in collisions occurring at various levels of BAC. Indeed they have contributed to the widespread adoption of limits lower than 0.8 g/L and by 2012 most countries, of what is now the European Union, had a limit of 0.5 g/L, many of them with lower limits for commercial or novice drivers, and with one exception the others had limits of 0.2 g/L or zero [10], the latter being a doubtfully enforced survivor of erstwhile Soviet influence. However, arguments are still being advanced for more public education, stricter enforcement, and further reduction of limits, supported by estimates that numbers of road deaths that are related to drink driving remain large. An estimate quoted widely in Europe is that up to 25% of road deaths in the European Union are so related, which stems from an extensive study for the European Commission in 2014 of drink driving in Europe which concluded that 20 to 28 % of all road deaths in the European Union in 2012 could be attributed to drink driving [3].

Ways of counting life-changing injuries on the roads that are comparable among countries have yet to be agreed. Doing the same for the deaths also has its difficulties, but European countries publish numbers of road deaths that are widely treated as comparable. These countries vary in the coverage of measurements of the BACs of drivers and other road users involved in fatal collisions, but this has not prevented countries from estimating how many of the deaths on their roads each year are “drink-related”. A starting point for achieving comparability among these estimates would be an agreed definition of a drink-related road death. An important step was made by the European research programme SafetyNet [14], which proposed a performance index implying the definition: death from a collision where any driver, rider, or pedestrian involved has a BAC above the legal limit. This definition has had widespread influence upon practice in European countries but has three serious shortcomings: the legal limit differs among jurisdictions, application of the definition requires more breath-testing in the immediate aftermath of collisions than is in some countries practicable or affordable, and the deaths that would have been less likely to occur if the relevant road users had drunk less are not confined to incidents where the legal limit has been exceeded. For example, the author has estimated [15] that in England and Wales in recent years for every four deaths recorded in collisions where the limit of 0.8 g/L has been exceeded there has been in addition one death in a collision involving a drink-driver without the limit being exceeded. It seems important to look for a definition which is independent of the legal limit, is widely applicable and includes all deaths that might have been avoided if the relevant road users had drunk less.

Lack of such a definition and thus of comparability among numbers of road deaths recorded as drink-related in different European countries presented a challenge to the ETSC in the programme PIN [16] which since 2006 has been ranking road safety performance in countries across the European Union and some neighbouring countries. This challenge was addressed in respect of progress in tackling drink driving in the first annual PIN report [17] under the author’s leadership by noting that although differences in recording of deaths as drink-related prevented direct comparison among the resulting numbers for different countries, it did not prevent comparison among the rates of change in these numbers over a period of years during which the criteria for designating a death as drink-related remained unchanged in each of the countries being compared. This comparison was made by estimating by log-linear regression for each country the average annual percentage change in the annual number of (a) road deaths recorded as drink-related and (b) other road deaths, and then ranking countries for progress in tackling drink driving according to the difference between the changes (a) and (b)—thus allowing for the fact that many measures taken to reduce road deaths in general also reduce drink-related deaths regardless of performance in tackling drink driving itself. This procedure was first carried out [17] for 20 countries over 9 years ending in 2005 and has been repeated with small refinements every few years, most recently for 23 countries over 9 years ending in 2018 [18] during which the definitions of a drink-related road death had remained as in Table 5 of [18]. The result is shown in Figure 3.

Each time this procedure was carried out it indicated faster reduction in drink-related deaths than in other road deaths in about two-thirds of the countries for which data were available, and an aggregate average reduction per annum in drink-related deaths over the period concerned over all the countries compared that was a percentage point or so greater than the corresponding reduction in other road deaths.

It seems therefore that efforts in Europe to address the challenge of drink driving, which consist mainly of the imposition and enforcement of legal limits on drivers’ BAC, associated public education and information and driver rehabilitation measures, are contributing if anything somewhat more than their share to the overall effort to reduce deaths on the roads. Against this background, some aspects of these efforts to reduce drink driving are next discussed.

## 3. Setting, Communicating and Enforcing Legal Limits on Drivers’ BAC

### 3.1. Setting the Limit

Against the background of the longstanding advice “Don’t drink and drive” the setting of a BAC limit above which it is illegal to drive can be seen as a matter of how far a specific law can best contribute to promoting a desired behaviour by a large proportion of the population. The findings from Grand Rapids stimulated a clearer recognition than hitherto that a BAC limit should be able do so, with limits in many countries at first being set at a level that would impinge mainly on a minority who are quite heavy drinkers and on them only when they have been drinking heavily. As such the setting of the limit need not have intruded greatly into the lifestyles of the majority—though its effect in reducing drink-driving may well have been fortuitously enhanced by many law-abiding moderate drinkers’ overestimating how close their drinking was bringing them to the legal limit. However, as countries have adopted lower limits these have become more intrusive until the lowest limits affect everyone except total abstainers quite a lot of the time. Nevertheless, where a BAC limit has been lowered, the lower limit has usually remained in force.

In the interests of respect for the law, it is important for a new or amended law to attract widespread compliance and be enforced in ways that maintain respect for it. A country is therefore wise to look for a clear balance of opinion in favour of lowering a BAC limit before doing so, and not to set a limit lower than it is practicable to enforce effectively. There are sound practical reasons why it is hard to enforce a limit lower than 0.2 g/L and these should be borne in mind where a limit of zero or so-called zero tolerance are advocated, but wherever the current limit is higher than 0.2 g/L, scope for reducing it should ideally be reviewed regularly.

### 3.2. Communicating the Limit

For a law to be effective, it is important for affected citizens to understand what the law requires of them and why. When a new law is imposed or an existing one is being changed, there is an onus on government to communicate this understanding and to encourage its acquisition by the affected citizens by use of appropriate means of public information. Government can be helped in this by relevant organisations who may well reach the ears and eyes of some citizens more readily than do channels of communication usually used by government. In the case of a BAC limit, these include manufacturers and suppliers of alcoholic drinks as well as road user and road safety organisations.

This task of communication is not just a once-for-all exercise at the time a BAC limit is introduced or altered; it is also an ongoing task because the drinking population, drinking culture and the range of available alcoholic drinks are continually evolving. Not only does each year a fresh cohort of young people become eligible to drive and, not necessarily at the same age, become eligible to buy alcoholic drinks, but also changing tastes and fashions in social life are continually altering the prevalence and kinds of social drinking among people of all ages and all kinds. Changes in the marketing of drinks of different kinds contribute to changes in taste and fashion, and also include changes in the alcohol content of familiar drinks in familiar-sized containers that make it advisable for drinkers to reassess how best to keep their BACs within their intended ranges. All this adds up to a continuing requirement for public information to help people who consume alcohol to keep within the BAC limit whenever they are driving over the course of day to day life. In European countries up to several decades of continually updated public information has contributed to very large majorities of their populations’ regarding drink-driving as socially unacceptable—not just those who strictly don’t drink and drive but also many of those whose approach is more flexible. 

Those who interpret the advice “Don’t drink and drive” flexibly face the question whether to equip themselves with and learn to use devices for measuring their own BAC. These same drivers also need understanding of how to estimate, after they have finished a session of drinking with a BAC over the limit, how long it will be before they are back within the limit and thus able once again to drive legally. For government, the existence of personal BAC measuring devices raises the questions how to regulate their availability, quality, and reliability and to advise on their use in the face of the dilemma that people with access to such devices may use them mainly to discover just how much they can drink without exceeding the limit.

### 3.3. Enforcing the Limit

For some laws regulating day to day behaviour, the law-abiding citizen hardly needs to be concerned about how they are enforced unless they are tempted to break them or suspected of breaking them or are unfortunately a victim of their being broken. However, the legal BAC limit is not one of these. It is a matter of life and death for all road users and it is broken daily in public by an appreciable minority of drivers, so a jurisdiction that imposes a BAC limit faces a public expectation of transparency as to what is being done to try to enforce the limit effectively.

The availability of portable evidential breath-testing devices makes extensive roadside breath-testing of drivers a practicable and understandable tool for visible enforcement, but this calls not only for careful definition of the requisite police powers having regard to social justice and civil liberties but also for the allocation policing resources upon which society places many other demands.

Effective enforcement has two distinct but closely related purposes: “deterrence” to discourage breaking of the law and “detection” to enable penalties to be imposed on those who do so and steps to be taken to discourage or prevent them from doing so again. Detection contributes to deterrence through reporting in local media of those who are convicted and the penalties they receive, which confirms both that offenders are being detected in the locality concerned and what levels of penalty are actually being imposed. Deterrence in turn relies upon public perception of a real likelihood that if one drives with a BAC above the limit then one may be detected and face conviction and the relevant penalty.

A principal means of detection is for drivers to be stopped and breath-tested at the roadside and to achieve deterrence for this to be believed to happen in such a way that anyone driving anywhere at any time thinks that it might be about to happen to them. This requires police or other authorised patrols both to have the power to require anyone driving or about to drive to take a test before proceeding with their journey and to be equipped to carry out the test reliably, preferably on the spot. Public acceptance of this power in a free society requires spelling out the circumstances in which the power may be used and how it should be exercised. In particular, it is unlikely to be acceptable for breath-testing to impinge disproportionately on certain kinds of people just because they belong to particular social groups.

Because patrols carrying out breath-tests is resource-intensive in terms of both staff-time and money; however, deployment should be deliberately intelligence-led. The general knowledge that a police force has of its area may well include a good deal of understanding of when and where drink driving is more prevalent or less so, and this understanding can be reinforced if there is a policy for breath-testing as many as is practicable of drivers, riders and pedestrians involved in collisions attended by the police or other authorised patrols. The pattern of occurrence of BACs above, say, 0.2 g/L in collisions gives useful indication not just of where and when drink driving is taking place but of where and when collisions are associated with it. Information of these kinds can contribute strongly to a rational basis for allocating available patrol effort—but it is advisable also to devote a small fraction of effort to patrolling at least likely places and times, so that tests may be seen really to be required anywhere and at any time. Once having decided to test at a particular time and place, it is then important for perception of fairness that the first and then each subsequent driver to be tested is chosen in a randomised way so that each driver who is around at the time has the same chance of being tested, irrespective of, for example, personal appearance or type and condition of vehicle.

Roadside breath testing, implemented in a variety of ways, is used extensively in European countries. Only about half of these keep national records of the level of testing, but for those that do so the ETSC has tried to follow the annual numbers of tests per thousand population since 2010 [19]. These differ widely, ranging from about 10 to about 600. The records also show the percentages of test results above the legal limit, which range from less than 1% to over 10%, with some tendency to be higher where the number of tests is lower. This would be consistent with the smaller numbers of tests being more targeted upon times and places where drink driving is more prevalent, but the correlation is not close. As part of wider international research into the attitudes of road users, car drivers in 20 European countries were asked in 2018 [20], quoted in [18] how likely they thought they were to be checked for drink driving on a typical journey. The proportion replying that they thought the probability was high or very high averaged 22.5% over these 20 countries, ranging from about 10% to over 50%, with little correlation except at the extremes with annual numbers of tests per thousand population across the 10 of these countries for which [19] provides this indicator. 

Table 4 of [18] shows that 26 European countries recorded numbers of drink-related deaths on their roads in 2017 or 2018, and comparison of these numbers with their corresponding total numbers of road deaths shows that the proportions of road deaths recorded as drink-related ranged from 1% to about 30%, and most are well below the 25% estimated for the European Commission in 2014 [3]. Such a wide range must stem in part from differences in the recording of drink-related deaths, which is understandable in part because breath-testing of those involved in collisions is resource-intensive and subject to practical difficulties. Differences in definition also contribute to the width of the range but this is from 5% to about 30% even over the 19 of these countries that can be regarded from Table 5 of [18] as working towards the SafetyNet definition. Nor do these percentages show any correlation with the annual numbers of drivers per thousand population checked for drink-driving in the 12 countries for which estimates of both figures are available.

This mixed picture indicates a need for research into the recording of drink-related road deaths and into the implementation and effectiveness of roadside breath-testing as a means of reducing drink driving in European countries. However, detection and drivers’ perception of the likelihood of detection are not the only important aspects of enforcement that contribute to deterrence; another is the level of penalties imposed on drivers convicted of exceeding the legal limit.

## 4. Treatment of Convicted Drink-Driving Offenders

Penalties imposed on convicted drink driving offenders in European countries for the offence of driving or being in charge of a vehicle while over the legal BAC limit range from modest fines or a few penalty points through higher fines and more penalty points to requirement to drive only a vehicle fitted with an alcohol interlock, disqualification, seizure of the vehicle, community service, or in extreme cases imprisonment. The level of penalty can depend on the severity and consequences of the offence and can be augmented if the driver is convicted of other offences such as dangerous driving committed at the same time as exceeding the BAC limit. The penalty can include or be abated by participation in a rehabilitation course or other measures to discourage reoffending.

However, imposition of a penalty upon conviction is only the beginning of treatment of these offenders. They will still be around for the rest of their lives and, except for a tiny minority while they are in prison and a few who choose to give up either drinking or driving, they will still be looking to mix choosing to drink and choosing to drive in their day to day living. In terms of the prospect of reoffending they range from feeling remorse or regret, and thus being open to help not to reoffend, at one end of a spectrum to, at the other end, being so affected by alcohol or by other medical conditions as to justify measures intended to result in their not driving at all.

Towards the first end of this spectrum, rehabilitation courses can help offenders to understand better how alcohol affects their behaviour, including their driving, and how to keep their BAC within the legal limit. In Great Britain, for example, a court convicting a driver for exceeding the limit can, if it thinks fit, offer the offender the opportunity to participate, if they wish, in an approved course, and thus reduce by up to a quarter their period of disqualification, which is a mandatory part of the penalty in Britain. A ten-year trial of this provision in a large national sample of courts in England and Wales estimated a halving of reoffending in the three years following conviction among those who took up the option compared with others offered the option in the same courts who chose not to take it up [21]. A Europe-wide study of rehabilitation courses for drink drivers other than problem drinkers also found a halving of reoffending to be achievable [22].

Towards the other end of the spectrum some European countries have defined a category of “high-risk offender”, comprising for example repeat offenders and those found to have driven with a BAC above some very high level or after drinking in combination with use of other recreational drugs. Such offenders can be subjected to extra requirements compared with first-time offenders convicted for exceeding the limit by less extreme margins—for example requiring satisfactory results of tests indicating how their use of alcohol affects their medical condition before their licence is restored after a period of disqualification.

Among the range of penalties short of imprisonment, disqualification probably has the greatest deterrent effect, at least among those who think in terms of complying with it if they became subject to it. However, disqualification does not prevent driving and for those whose driving would not attract the attention of the police, driving while disqualified may incur little risk of detection. So for those who are prepared to drive while exceeding the BAC limit, disqualification may do little to discourage reoffending. This limitation is addressed by technology in the form of the “alcohol interlock”, which for as long as it is fitted to the car most used by the offender and is in working order actually prevents reoffending, at least in the form of the offender driving the fitted car.

## 5. Alcohol Interlocks and Their Use in Europe

An alcohol interlock fitted to a motor vehicle is a device which enables an intending driver to provide a breath sample, estimates a BAC from the sample and allows the vehicle to be started only if the BAC is below a certain limit. When the vehicle is being driven, the device from time to time indicates to the driver that the engine will be turned off unless a further satisfactory breath sample is provided within a stated time. The driver then has that length of time in which to find a safe place to stop and provide the sample in order to continue their journey. The upshot is that the vehicle can only be used by drivers with a BAC within a limit specified in the device. The device can also keep a record of its use and the BAC levels estimated from the breath samples. Readiness for fitting with the device has hitherto differed among various makes and models of vehicles, but revision in 2019 [23] of the General Safety Regulation concerning type-approval of motor vehicles in the European Union should result in all new vehicles there being similarly ready to equip with alcohol interlocks from early in the 2020s.

There are two main kinds of use for alcohol interlocks: voluntary use, in which the vehicle owner has reason to guard against the vehicle being driven by anyone exceeding some BAC limit and fits the device to achieve this, and mandatory fitting and use in the context of enforcement of a legal BAC limit or of a requirement that vehicles used for a certain purpose be fitted.

Examples of voluntary use are fitting by commercial operators who wish to satisfy themselves, and possibly the public, that their drivers will be within a certain BAC limit, and fitting by a household to a household car where it is known that one or more of the household members may be tempted to drive after drinking too much. Voluntary use is simply a private matter of the owner having the device fitted by a supplier and arranging for its maintenance, learning to use it, and learning to use the record that the device can keep of attempts to provide satisfactory breath samples.

Mandatory fitting includes fitting to the main vehicle used by a convicted drink-driving offender to prevent reoffending in that vehicle for a certain period, or fitting by a commercial operator to a vehicle or fleet of vehicles to enable use for a purpose for which fitting is legally required, such as carriage of children on school journeys. Mandatory fitting is necessarily more complex than just acquiring and maintaining the device. There has to be legislation to make the fitting and use mandatory, as is permitted to countries of the European Union under its Driving Licence Directive. Legislation usually provides for monitoring of use, which in turn generates data whose handling has to meet data protection requirements. Agencies of law enforcement and criminal justice each have their parts to play.

As an addition to the range of penalties available to courts in sentencing, the alcohol interlock offers the advantages over disqualification that it may allow the offender to continue in their occupation, and thus support dependents, and it may be more effective in avoiding reoffending, but sentencing to impose use of an interlock requires the cost of fitting and operation to be met. It is advisable for its imposition to be accompanied by rehabilitation measures because research has shown that otherwise its effect on the offender’s driving behaviour may well not persist beyond the required period of fitting to the vehicle they mainly use. This use of alcohol interlocks is longstanding in North America, where, for example. deaths in drink-related collisions were found in a study covering 2004–2013 [24] to be about eight per cent fewer in 18 states of the USA that had made interlocks mandatory for convicted drink-drivers than in 32 states without such a requirement. In Europe since the turn of the century experience has been gained gradually as use has become progressively more widespread. The ETSC has monitored this process, including aspects of effectiveness in reducing reoffending, with the help of case studies from various countries, and has offered guidelines to countries which may be contemplating adopting the alcohol interlock as a penalty [25]. Since 2008 the ETSC has kept track of the process in the form of an “Alcohol Interlock Barometer” which features a map showing stages that European countries have reached in using the alcohol interlock in a website providing a range of relevant information country by country [26]. Figure 4 shows the maps for 2008 and 2015, in which the colour blue indicates countries that had reported no use so far and other colours indicate the stage reached by each country. The differences between these two years exemplify the spreading use of the alcohol interlock, which has since continued.

To complement the Barometer, the ETSC published late in 2020 a detailed inventory [27] of the seven national alcohol interlock programmes for convicted offenders then current in Europe, with information about an eighth just then getting under way and steps towards introduction in three other countries.

Issues that have arisen as various countries with their differing legal and judicial systems have explored the mandatory use of alcohol interlocks have rarely been with the technology as such but have included readiness of the judiciary to make use of alcohol interlocks as part of penalties, relationship of requirement to have an interlock fitted to measures to rehabilitate the offender, the cost of the devices and support systems and who pays (for example how acceptable it is for the choice of penalty available to the court to depend on the means of the offender), and use of data recorded by the devices to provide support to the offender and to inform policy.

## 6. Relevance of Experience of Managing Drink Driving to Management of Other Drug Driving

The challenge presented to road safety by other recreational drugs is similar to that presented by alcohol in that they impair capacity to drive safely so that the sensations that they offer to their users are hard to combine with safely of driving. However, there are many practical differences in management of the challenges. The author has no claim to expertise in relation to other drug driving but offers the following comments.

Widespread use of alcohol is very longstanding and its main forms and sources of supply to consumers are also longstanding and are regulated in ways that are familiar to its users. This was the case before the issue of effects on widespread driving of motor vehicles came to be recognised and provided a familiar background against which the challenge to road safety has been addressed. Many of the other recreational drugs, in contrast, have come into really widespread use only in recent decades, the ranges of substances concerned and patterns of use are wide and quite rapidly evolving, the possession, supply and use of most of them are subject to strict legislation and channels of supply are strongly influenced by criminal activity.

In terms of managing effects of drink driving on road safety, a lot had already been done to deal with alcohol before the use of other recreational drugs became widespread, and this provides experience that can be drawn upon in managing effects of other drug driving, but in doing so it should be borne in mind that management of drink driving is helped by several circumstances peculiar to alcohol. Alcohol starts to impair driving soon after it enters the bloodstream, the level of impairment is related in known ways to the BAC level, impairment lasts for as long as there is still appreciable alcohol in the blood but no longer, the BAC level can be estimated reliably and non-invasively by breath-testing, the businesses of producing and supplying alcoholic drinks are highly organised and disposed to help society in mitigating adverse effects like those on road safety, and the range and characteristics of drinks on the market evolve relatively slowly.

In contrast, the range of other recreational drugs and their biochemical properties are evolving rapidly, their producers and distributors are hard to involve in managing drug driving, some of the drugs remain in the body for much longer than the impairment they cause lasts, and not for all of them is there good understanding of the relationship between their level in the body and the resulting level of impairment or how to detect and measure their presence and whether they are causing impairment at the time. All this makes it more difficult than for alcohol to set, keep up to date, communicate and enforce limits on their presence in the body above which driving can be made illegal.

Nevertheless many European countries have introduced legislation, public information and rehabilitation and health care aimed at addressing drug driving. Alongside the use of recreational drugs these can relate also to medication that affects driving, but whereas for medication it may be practicable to determine medically safe levels of use when driving, this is harder to contemplate, let alone implement, for substances whose possession, supply and use are widely illegal. The ETSC has summarised [28] various approaches being adopted in European countries and ways in which they can be used to help tackle drug driving in the context of other underlying issues related to drug use.

## 7. Discussion

European countries are in their fifth decade of managing drink driving in ways that focus on reducing that part of the burden of death injury and damage on the roads that would be prevented if drivers drank less before driving or drove less after drinking. They have come to be doing this mainly by imposing, enforcing, and tending to lower legal limits on drivers’ BAC, with associated ongoing public education and information and driver rehabilitation measures. These efforts are associated with contributing if anything somewhat more than their share in recent years to the overall reduction in deaths on the roads of Europe, and in terms of public attitude, a large majority of people now regard drink driving as socially unacceptable. The latter is a substantial and valuable shift from the days before BAC limits when those opposed to them could argue that people drove better after a few drinks. However, the signs are that up to about a quarter of road deaths are still drink-related, which taken with the changing nature of drinking culture and in the driving population whose behaviour needs to be influenced means that work remains to be done.

One starting point is to characterise the driving population as comprising two broad groups: those who largely succeed in living by an intention to follow strictly the by now classic advice, “Don’t drink and drive”, and those who more flexibly mix their choosing to drive with their choosing to drink alcohol. Those in the first group rarely have a collision that might not have happened if they had drunk less so almost all drink-related collisions involve drivers in the second group. Some drivers may move from one group to the other over their life-cycles, and if the first group grows as a result, then road safety should benefit, but it makes sense to suppose that most of the second group will go on as they are. Moreover, the majority of the population who regard drink driving as socially unacceptable is so large that it must include many of this second group, who can thus be supposed to share the widespread understanding that driving after drinking is risky and to wish to keep this risk down. These drivers need help to make choices that reduce or at least moderate their drink driving and at the very least to comply with the legal BAC limit—help through information and encouragement, and if they are convicted for a detected lapse, through rehabilitation as part of or alongside the penalty the law requires. There remain a minority of the second group who seem to share neither the widespread view that drink driving is socially unacceptable and risky nor respect for the law, and to be ready to drive with high or very high BACs and do so again after being convicted and penalised for the offence. As indicated for example in a recent report taking stock of management of drink driving in the UK [29], the interests of road safety call for such drivers to be supported in changing their harmful behaviour where there are medical or psychological reasons for it, or otherwise to be persuaded to change it. Failure to achieve such change leads to a requirement for them to be legally restrained from driving.

A jurisdiction reviewing drink driving legislation for its own people having their own variant of these characteristics, even though well informed about experience across Europe, may well find itself lacking in relevant knowledge. In one respect, namely whether enough of its people are ready for a BAC limit to be imposed or an existing limit higher than 0.2 g/L to be reduced, they can find out by commissioning competent surveys of opinion. However, in other important respects, European experience does not yet provide the knowledge the jurisdiction would like to have to guide it.

The jurisdiction would like to estimate how much harm drink driving is causing in its territory through drink-related collisions and how much this could be reduced by legislative measures that it is contemplating, but there is as yet no comprehensive definition of drink-related collisions for this purpose—no definition which is independent of the legal limit, is widely applicable and includes all collisions that might have been avoided if the relevant road users had drunk less. There is a body of evidence about the proportion of those collisions occurring at a certain BAC level that would be prevented if the BAC level were reduced by a certain amount, but evidence is lacking about the reductions in drivers’ BAC levels that would result from particular legislative measures being imposed. Likewise in terms of enforcement of BAC limits by roadside breath-testing, evidence is lacking about how much drink driving in an area can be reduced typically by a certain expenditure on breath-testing or how best to deploy those resources to maximise the achieved reduction in drink driving.

## 8. Conclusions

To build upon Europe’s half century of progress in managing drink driving thus requires further advances in both research and practice. Better measurement of the burden of road casualties stemming from drink driving requires research into the identification and recording of drink-related collisions. More effective enforcement of drink driving law requires research into drivers’ response in terms of drinking and driving to the deployment of various measures to discourage them from drink-driving or at least to reduce the BACs at which they drive. The latter research would best be embarked upon not with the expectation of finding tidy numerical answers to questions that can be formulated only in rather imprecise terms, but in search of soundly based broad indications to which those making and implementing legislation could look for guidance.

In terms of practice, research in these challenging areas should in no way delay continual sharing of and learning from experience in the strategy, tactics, and operational practice of balancing deterrence and detection in enforcement of drink driving law, in communication of the law to drinkers as their tastes and habits and the range of drinks available to them evolve, and in the use of alcohol interlocks as a penalty accompanied by rehabilitation measures. Nor should there be delay to available steps to make the vehicle fleet simpler to fit with alcohol interlocks when they are needed or to make their fitting and use less expensive, so that they become more readily available to courts for use in sentencing and to voluntary users. There may also be a role for driver assistance devices that can detect inattention reliably and induce an affected driver to take a break. In relation to reducing harmful driving behaviour, there is scope for more supportive trained help enabling people with alcohol problems or other behaviour-influencing health issues to live safely with these. In ways like these and with due determination there is good reason to look for further reduction in drink-related deaths and injury on the roads of Europe.

## Figures and Tables

**Figure 1 ijerph-17-09521-f001:**
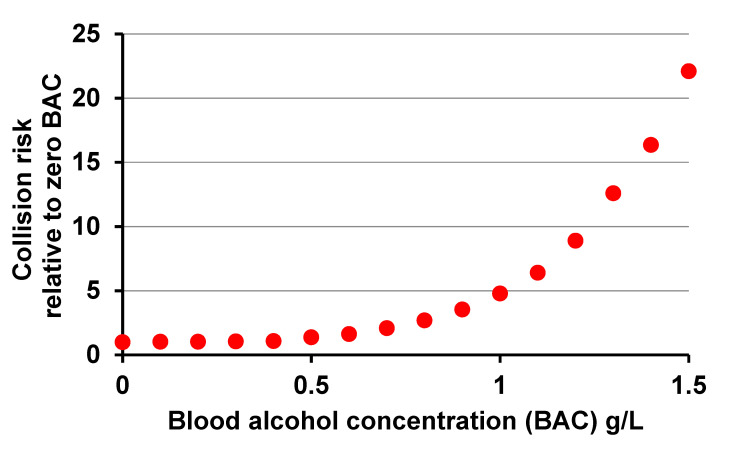
Variation in the risk of involvement in a collision, however slight, with level of alcohol in the blood as estimated in a repeat of the Grand Rapids study by Compton et al. [8].

**Figure 2 ijerph-17-09521-f002:**
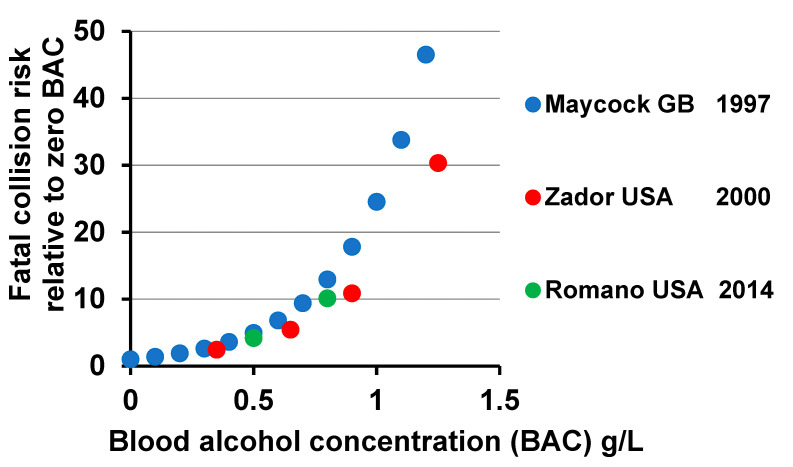
Variation in the risk of involvement in a fatal collision with level of alcohol in the blood as estimated by Maycock [11], Zador et al. [12] and Romano et al. [13].

**Figure 3 ijerph-17-09521-f003:**
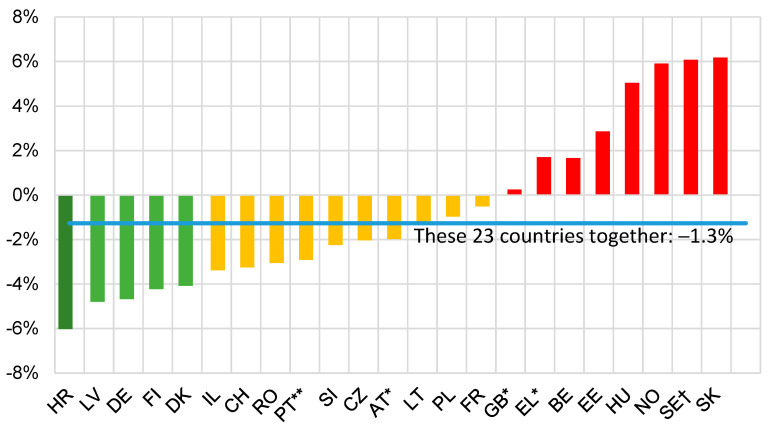
Reproduced with permission from the European Transport Safety Council. Difference between the average annual percentage change in the number of drink-related road deaths and the corresponding change for other road deaths over the period 2010–2018 in 23 countries [18]. A negative difference indicates faster reduction in drink-related deaths. * 2010–2017; ** 2010–2015; ^†^ driver deaths only; Key to country codes: HR—Croatia; LV—Latvia; DE—Germany; FI—Finland; DK—Denmark; IL—Israel; CH—Switzerland; RO—Romania; PT—Portugal; SI—Slovenia; CZ—Czech Republic AT—Austria; LT—Lithuania; PL—Poland; FR—France; GB—Great Britain; EL—Greece; BE—Belgium; EE—Estonia; HU—Hungary; NO—Norway; SE—Sweden; SK—Slovakia.

**Figure 4 ijerph-17-09521-f004:**
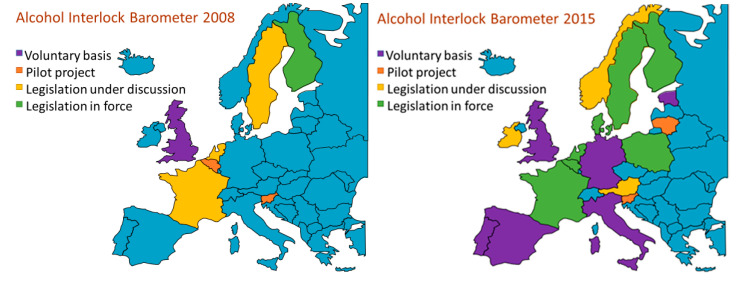
Reproduced with permission from the European Transport Safety Council. Extent of use of alcohol interlocks in European countries in 2008 and 2015 as reported to the ETSC at the time—colour blue indicates no use reported so far.

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
