# Peer review of "Drink Driving as the Commonest Drug Driving—A Perspective from Europe"

_ijerph, 2020, doi:10.3390/ijerph17249521_

Round 1

Reviewer 1 Report

This manuscript comprises an interesting review about the relationship between the risk of mixing the driving of motor vehicles with the consumption of alcohol.

However, in my opinion some minor concerns must be attended:

In figure 3, the abbreviatures used to indicate data from different countries are clear for European citizen, but not for citizen from other countries. Complete country names must be included. Also a legend like “Figure reproduced with permission from European Transport Safety Council” must be included.

In Figure 4, author has more recent data (at least 2017). Then, why author chose to compare 2008 data with 2015 data?

Discussion section. Some recent papers studied potential lives saved by in vehicle-alcohol detection systems considering several levels of BAC:

Charles M. Farmer (2020) Potential lives saved by in-vehicle alcohol detection systems, Traffic Injury Prevention, DOI: 10.1080/15389588.2020.1836366

Patrick M. Carter, MD, Carol A. C. Flannagan, PhD, C. Raymond Bingham, PhD, Rebecca M. Cunningham, MD, and Jonathan D. Rupp, Modeling the Injury Prevention Impact of Mandatory Alcohol Ignition Interlock Installation in All New US Vehicles. Am J Public Health. 2015;105:1028–1035. doi:10.2105/AJPH.2014.302445

I suggest to consider these papers in this manuscript.

Minor comments:

Some additional references that probably can help to improve the discussion  are the following

Lund AK, McCartt AT, Farmer CM. 2007. Contribution of alcoholimpaired

driving to motor vehicle crash deaths in 2005. Proceedings

of the 18th international conference on alcohol, drugs, and traffic

safety (CD-ROM). Oslo (Norway): International Council on

Alcohol, Drugs, and Traffic Safety

Lund AK, McCartt AT, Farmer CM. 2012. Contribution of alcoholimpaired

driving to motor vehicle crash deaths in 2010. Arlington

(VA): Insurance Institute for Highway Safety.

Voas RB, Torres P, Romano E, Lacey JH. 2012. Alcohol-related risk of

driver fatalities: an update using 2007 data. J Stud Alcohol Drugs.

73(3):341–350. doi:10.15288/jsad.2012.73.341

McCartt AT, Wells JK, Teoh ER. 2010. Attitudes toward in-vehicle

advanced alcohol detection technology. Traffic Inj Prev. 11(2):

156–164. doi:10.1080/15389580903515419

Author Response

Thank you for the attention you have given to my submission

In the captions to Figures 3 and 4 I have inserted permission from ETSC to reproduce them, and in the caption to Figure 3 I have added a key to the country codes.  I have preferred doing this to editing the axis in the chart to show the full names of the countries because this would have separated the substance of the chart from the caption - and not everyone is comfortable reading diagonal or vertical text.   Thank you for pointing out the need for this

In Figure 4 my aim is to show what progress had been made by the middle of the decade and encourage those who are interested to visit the Barometer to find an update on continued progress to whenever they are reading.   Those who do so should detect acceleration of progress and that the process has not been without a setback.   Below Figure 4 I have inserted mention of a new ETSC publication which provides an overview of interlock progress, giving much fuller information as at late 2020 than the Barometer. The latter will continue to be updated regularly

Thank you for the references.   Seeking out Patrick et al led me to an even more relevant paper from the same journal which I have cited at lines 550-557 to support my mentioning the longstanding use of interlocks in North America.   I have already cited under the name of Romano a later account of the work of Voas et al.  

Reviewer 2 Report

Dear author,

A nice review of the subject with clear language.

-Abstract could be improved; with a clearer structure making it more intuitive for the reader what is known, what is the aim of this paper and what are the conclusions and recommendations of the paper;
- Avoid such long sentences. For example, the first sentence of the abstract is long and is phrased in a way that can be confusing. Maybe try to rephrase it to make it an easier and more fluid read. Also avoid using so much the passive voice: in line 11 when explaining the aim of the paper, again make it clearer and if you rephrase it to make it clearer you can avoid the passive voice all the time;
- Despite the author's knowledge of the topic, the introduction could benefit from a clearer explanation of what is known backed-up with references. For example, in line 41 you state alcohol consumption is high in european countries however you offer no reference to back up that claim. Facts stated should be supported by appropriate references;
- Another suggestion for improvement is to better justify your focus on the E.U. not just from the high consumption but also from the impact of that consumption: make it clearer why this is a public health problem and why it is a public health problem in European countries;
- Associated with my previous comment I would add that it would be important for you to frame the topic: what is the relevance/burden associated with the topic? Back it up with appropriate references. In section 2 you explain a bit more the magnitude of the problem and later on develop the issues of different measurements in different countries. Despite this, I believe you can give a broader picture earlier in the introduction and focusing more on the health impacts: morbidity and mortality of a topic you are framing as a public health problem;
- In the 3rd paragraph of the introduction you explain a bit your methodological approach: can you explain better? You mention your previous experience, the lecture you have given and that you based yourself on some ETSC reports. How did you go about this? Did you have any specific approach to the literature review?
- In the last paragraph of the introduction you explain a bit the structure of the article. Maybe this paragraph could be improved in order to make it easier for the reader to understand your sections and know the structure and questions answered throughout the rest of the article. Also explain clearly here the aim of this article.
- Sentence in line 458-459: any reference to back up this claim?
- Line 479: please rephrase this. The subject of your sentence is not clear.
- In the second paragraph of your conclusion you briefly mention the need to inform and encourage some drivers. Given the scope of his journal I would have liked to see this further explored. From a public health perspective what could be done? What interventions could be effective? What evidence exists already? How can this article inform public health teams developing interventions and/or policies?

Overall, I would recommend: i) making the sentences more fluid and shorter; ii) keeping a public health lenses throughout the article which I believe at points you don't have and for me is the bigger issue of the article; iii) make sure you use appropriate references.

Author Response

Thank you for the attention you have given to my submission and your helpful comments and suggestions concerning the abstract and introduction.

I wasn't that happy with the Abstract so I have had another go at it helped by your advice,

I have also quite substantially reshaped the Introduction, trying to take account of your very helpful advice.   This has introduced two new references (one of them to alcohol consumption data and the other to mortality data) and brought forward some of the existing references.

Lines 458-459 - now lines 645-646   This is a logical consequence of the definitions of the two groups in the previous sentence and has no other source.   I have sharpened the definitions to strengthen the logic.

Line 479 - now line 675   Thank you for spotting this.  'It' was a survivor from an earlier stage in drafting.

The emphasis in the concluding discussion on the legislative and regulatory processes follows from the emphasis of the whole perspective and what I feel qualified to write about.   I hope I have a reasonable awareness of the important public health dimension but I am not qualified to write in detail about it.   I have tried to add explicit mentions of aspects of it in the opening and final paragraphs of the concluding discussion, the former supported by a reference.

Reviewer 3 Report

Dear author,

I found your commentary interesting and of course the topic is of great importance. I was wondering if you could maybe include some information about the influence of legal drinking age laws on drink driving?

There are many onsets to prevent drink driving, which the author of the commentary discusses. In my opinion the effect of a minimum legal drinking age has massive influence for the prevention of drink driving (also see some literature below). I was wondering if the inclusion of this additional aspect to the commentary would complete the commentary and distinguish the manuscript from a summary of the examined articles. 

"The effects of minimum legal drinking age 21 laws on alcohol-related driving in the United States"   "The fatal toll of driving to drink: The effect of minimum legal drinking age evasion on traffic fatalities"   "The minimum Legal Drinking Age and Public Health"

Author Response

Thank you for the thought you have given to my submission.

Your suggestion for extending its coverage to include the relevance of the age at which it becomes legal for a young person to drink alcohol is a valuable one.   I am aware that this issue is of greater concern in the USA than I perceive it to be across Europe, where it differs among countries.

The setting of and compliance with this age is clearly relevant to the management of drink driving, and your suggestions for relevant papers has led me to Carpenter and Dobkin's widely cited paper from 2011, to which I am glad to have been introduced.   But I am not aware of consideration of the possible use of this legal age in managing drink driving in the European context.

I have looked but have not been able to see how I could address the subject in a satisfying way in the context of minor revisions to my commentary.  To raise this possibility in the European context on the basis of experience with changes in the age over the years in the USA seems to me to require a whole short paper in its own right, and for me to write that would require me to devote a good deal of time to relevant literature from the USA as well as to the form the age limit takes across Europe and how its relevance to drink driving has been considered in European countries.

I have therefore decided not to take up your valued suggestion in this commentary.